# Enzymatic enhancing of triplet–triplet annihilation upconversion by breaking oxygen quenching for background-free biological sensing

Ling Huang[1], Timmy Le [1], Kai Huang [1] & Gang Han [1]✉

Triplet-triplet annihilation upconversion nanoparticles have attracted considerable interest due to their promises in organic chemistry, solar energy harvesting and several biological applications. However, triplet-triplet annihilation upconversion in aqueous solutions is challenging due to sensitivity to oxygen, hindering its biological applications under ambient atmosphere. Herein, we report a simple enzymatic strategy to overcome oxygen-induced triplet-triplet annihilation upconversion quenching. This strategy stems from a glucose oxidase catalyzed glucose oxidation reaction, which enables rapid oxygen depletion to turn on upconversion in the aqueous solution. Furthermore, self-standing upconversion biological sensors of such nanoparticles are developed to detect glucose and measure the activity of enzymes related to glucose metabolism in a highly specific, sensitive and background-free manner. This study not only overcomes the key roadblock for applications of triplet-triplet annihilation upconversion nanoparticles in aqueous solutions, it also establishes the proof-of-concept to develop triplet-triplet annihilation upconversion nanoparticles as background free self-standing biological sensors.

[1] Department of Biochemistry and Molecular Pharmacology, University of Massachusetts Medical School, Worcester, MA, United States.
✉email: Gang.Han@umassmed.edu

Triplet–triplet annihilation upconversion nanoparticle (TTA-UCNP) has been emerging as the next generation of photon upconversion materials[1–8]. These nanoparticles hold great promise in a wide variety of important areas, such as bioimaging[9,10], therapy[11], photoredox catalysis[12] and solar energy harvesting[13], because of their unique properties such as high upconversion quantum yield, the need for low excitation power long-wavelength light (<200 mW/cm²) and potentially tunable excitation/emission wavelength[1–8]. In particular, optical triplet–triplet annihilation upconversion (TTA-UC) converts low-energy long-wavelength photon to high-energy shorter wavelength one[1–8]. During the process of TTA-UC, low-energy photons are first absorbed by a photosensitizer (Sen), which then reach its single excited state ($^1Sen^*$). Subsequently, the inter-system crossing process occurs. The energy of the photo-sensitizers transited to their triplet excited state ($^3Sen^*$)[1–8]. Due to the long-lived lifetime of triplet excited state, the energy of $^3Sen^*$ can transfer to annihilator (An) by triplet–triplet energy transfer process (TTET). Finally, two triplet excited state annihilators ($^3An^*$) undergo a TTA process to generate one high-energy singlet excited state annihilator ($^1An^*$), and subsequent short-wavelength light emissions (Fig. 1a)[1–8].

To date, a few TTA-UC photosensitizers and annihilators pairs with solid upconversion quantum yields have been reported in deaerated organic solvents[3,14,15]. Unfortunately, oxygen molecules diffused in water can rapidly quench $^3Sen^*$ and $^3An^*$. In particular, $^3Sen^*$ can rapidly sensitize oxygen molecules to generate singlet oxygen ($^1O_2$), thereby inhibiting the TTET process from $^3Sen^*$ to $^3An^*$, leading to the failure of TTA-UC (Fig. 1a). Therefore, although TTA-UC can be improved in an organic solvent by the approaches such as oxygen consuming[16], enhancing upconversion efficiency of TTA-UCNP in aqueous solutions has been challenging under an ambient atmosphere[1–8]. This oxygen-induced quenching problem is an especially strong obstacle in regard to biological applications that must happen in aqueous solutions under an ambient atmosphere[17–19]. In order to address this key problem, several methods have been attempted in recent years[10,20–26]. For example, TTA-UC dyes were

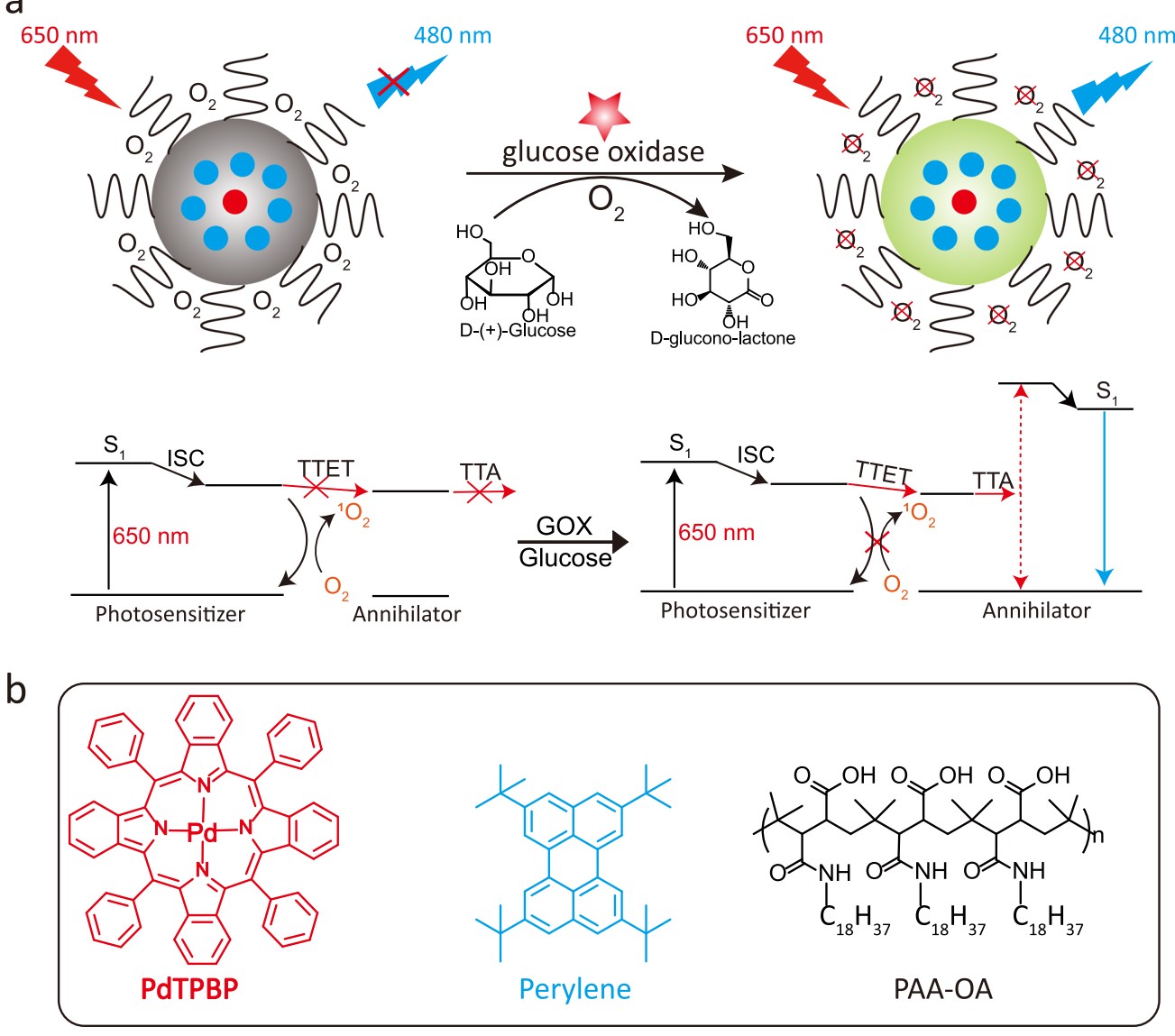

**Fig. 1 Solving the oxygen-quenching issue of TTA-UCNP with glucose oxidase. a** Schematic illustration and the mechanism of lighting up TTA-UCNP in the presence of glucose and glucose oxidase (GOX). The blue dot represents the annihilator of perylene, the red dot represents the photosensitizer of PdTPBP, the left gray nanoparticles stand for origin TTA-UCNP and the right green nanoparticles stand for TTA-UCNP in the presence of GOX and glucose. **b** Molecular structures of the photosensitizer PdTPBP, the annihilator perylene, and the amphiphilic polymer PAA-OA.

encapsulated in mesoporous silica to slow down oxygen diffusion and the concomitant oxygen quenching[10,20]. However, the existing silica TTA-UCNP cannot block the oxygen quenching completely and thus has suboptimal upconversion performance and the size of these nanoparticles is very large, typically over 200 nm[10,20]. Liposomes were also used to encapsulate TTA-UC dyes in order to construct water-dispersible nanoparticles[21–23]. Similarly, this method was also unable to effectively resolve the oxygen issue in an aqueous solution[21–23]. Recently, to reduce the oxygen-induced TTA-UC quenching, soybean oil has been explored to form TTA-UC oil droplets[24–26]. Yet, because of the quick oil/water phase separation, the resultant TTA-UC oil droplet appears to have difficulty staying in an aqueous solution. In addition, these TTA-UC oil droplets are also large, typically over 100 nm[24–26]. Therefore, developing a simple strategy to overcome oxygen-quenching trouble for stable water-dispersible and small-size TTA-UCNP under an ambient atmosphere is both urgently needed and highly desirable.

On the other hand, due to their unique optical and chemical properties, the development of TTA-UCNPs as biological sensors has been long-sought-after[26]. Organic TTA-UCNPs based sensors are expected to not only inherit these general merits from inorganic lanthanide UCNPs (e.g., minimized autofluorescence background interference, reduced light scattering, and photodamage), they also have certain distinctive advantages over their inorganic counterparts in regard to biological sensing applications[27,28]. Firstly, the emission of lanthanide UCNPs comes from a number of rare-earth ions buried and shielded inside nanoparticles, which cannot respond and react directly to analytes in an aqueous solution[27,28]. In order to be used as biosensors, such inorganic UCNPs typically have to be conjugated with other fluorophore-containing molecules (e.g., a cyanine dye, rhodamine dye, and ruthenium complexes)[29–31] or by growing materials whose absorption overlaps with upconversion emissions[32,33]. Yet, unlike conventional fluorescence molecule-based sensors, these inorganic UCNPs sensors rely on an inefficient luminescence resonance energy transfer mechanism or on emission-reabsorption process[34]. The readouts of such indirect sensors are often difficult to control due to variations regarding the amount and aggregation of conjugated dyes and the resultant materials[29–34]. In particular, completely quenching these multiple inorganic ion mediated upconversion emissions to create self-standing biosensors has been rather challenging[29–34]. Moreover, the emissions of lanthanide UCNPs are typically a mixture of multiple $f$–$f$ orbital transitions[27,28]. Since the ratios and intensities of these upconversion peaks and overall upconversion colors are heavily dependent on the power density of the excitation light and dopant amounts, the accuracy, sensitivity, and robustness of such sensors can often be compromised by the slight variation in these experimental factors[35]. In contrast, the emission of organic TTA-UCNP is based on the triplet excited states of photosensitizers and annihilators[1–8]. Thus, similar to organic fluorescence molecules, there is typically one emission peak and no color mixing problem in organic TTA-UC. Unlike inorganic UCNPs, organic TTA-UCNP can potentially be used as self-standing biosensors to detect analytes that can regulate and affect the triplet properties without any further nanoparticle modifications. In addition, compared to the weak absorption of lanthanides ions in inorganic UCNPs, TTA-UCNP utilizes intense long-wavelength absorbing organic photosensitizers, which lead to a much higher level of brightness and can be triggered with low power LED light[1–8].

It is well-known that glucose oxidase (GOX)-catalyzed glucose oxidation reactions can consume oxygen[36–38] (Fig. 1a). More importantly, GOX is biocompatible and has been widely used in medicine[39] and biology[40,41]. We envisioned that such a reaction

can be used to remove oxygen in an aqueous solution to turn on and amplify the upconversion emissions of TTA-UCNP (Fig. 1a).

Herein, we report on a simple and straightforward GOX-catalytic method that can quickly deplete oxygen and thus amplify the upconversion intensity of sub-50 nm TTA-UCNP in an aqueous solution. Furthermore, self-standing "turn-on" upconversion biological sensors are developed based on such enzymatic TTA-UC enhancement strategy. We demonstrated that TTA-UCNP can detect glucose and measure the activity of enzymes related to glucose metabolism in a highly specific, sensitive, and background-free manner. Thus, this study not only provides a straightforward method to overcome a key roadblock in using small-size TTA-UCNP for photonic and biophotonic applications in aqueous solutions but also establishes the proof-of-concept to develop TTA-UCNP as background free self-standing biological sensors.

## Results

**Synthesis and characterization of water-dispersible TTA-UCNP.** As a proof-of-principle, palladium(II) meso-tetraphenyl-tetrabenzoporphyrin (PdTPBP) and perylene were used as the sensitizer and annihilator, as they are one of the most investigated and effective long-wavelength activating TTA-UC dye pairs used in organic solvent[42]. As the annihilator, perylene has (Supplementary Fig. 1) high fluorescence quantum yield ($\Phi_f = 98\%$) and robust photostability[2]. PdTPBP is a potent far-red light-absorbing photosensitizer and its maximum absorption peak is located at 628 nm (molar extinction coefficient, $\varepsilon = 1.1 \times 10^5\,M^{-1}\,cm^{-1}$) (Supplementary Fig. 1). Moreover, the triplet excited state energy level ($T_1$) of perylene is 1.52 eV, which is energetically compatible with that of PdTPBP (1.55 eV)[2]. In our study, the optimal concentrations of PdTPBP and perylene are determined to be 10 μM (PdTPBP) and 250 μM (perylene) (Supplementary Fig. 2). Intense far-red to blue TTA upconversion emission can be observed with upconversion quantum yield up to 7.9% in deaerated toluene (Supplementary Fig. 3). The TTET rate between PdTPBP and perylene was found to be high ($k_{sv} = 2.0 \times 10^5\,M^{-1}$), as determined via Stern–Volmer quenching curves (Supplementary Fig. 4). The power-dependent TTA-UC spectra were further tested and the TTA-UC threshold intensity is observed to be 88.5 mW cm$^{-2}$ (Supplementary Fig. 5).

To obtain a water-dispersible TTA-UCNP, we encapsulated this TTA dye pair (PdTPBP/perylene) (Supplementary Fig. 6) in an amphiphilic polymer-polyacrylic acid substituted octadecyla-mine (PAA-OA). The size of the TTA-UCNP was estimated by transmission electron microscopy (TEM) to be 40.2 ± 3.2 nm (Supplementary Fig. 7). The hydrodynamic diameter of the TTA-UCNP was determined to be 57.3 ± 3.9 nm using dynamic light scattering. We also measured the UV–Vis absorption spectra of the TTA-UCNP. As shown in Supplementary Fig. 8, the TTA-UCNP presents the characteristic absorption peak from perylene at 440 nm and the characteristic absorption peak of PdTPBP at 628 nm (Supplementary Fig. 1), suggesting that the TTA-UCNP encapsulated both the photosensitizer and annihilator[43]. The entrapment efficiency of the TTA-UCNP is determined to be 92% (PdTPBP) and 75% (perylene).

Next, the photophysical properties of this TTA-UCNP in phosphate-buffered saline (PBS) were investigated. The schematic diagram of the instrument setup was presented in Supplementary Fig. 9. As expected, in the absence of GOX or glucose, the upconversion emissions of nanoparticles in PBS buffer were not detectable. In contrast, in the presence of both GOX and glucose, intense blue upconversion emission was observed at 480 nm (Fig. 2a). Using the well-established literature reported method,

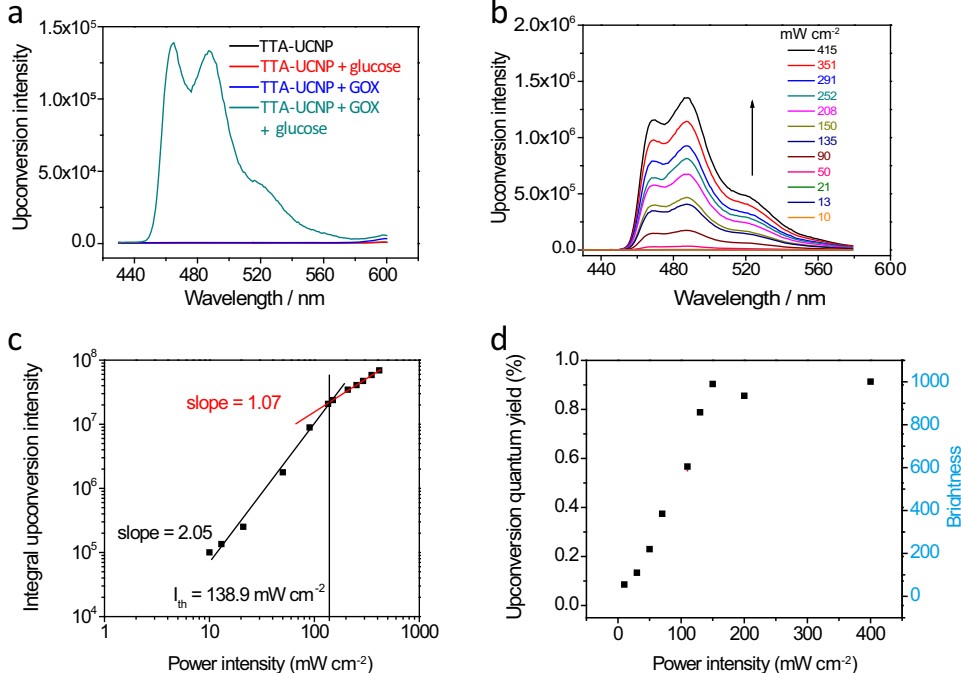

**Fig. 2 Upconversion performance of TTA-UCNP in different circumstances. a** TTA-upconversion spectra of TTA-UCNP at different conditions (TTA-UCNP only, TTA-UCNP with glucose, or with GOX, or with both glucose and GOX) in PBS buffer, $\lambda_{ex}$ = 650 nm. **b** Incident light power dependence study of TTA-upconversion emission intensity for TTA-UCNP in PBS buffer. **c** Double-logarithmic plot of perylene integrated emission intensity as a function of 650 nm excitation power density. Solid lines illustrate a slope of 2.05 (black, quadratic) and a slope of 1.07 (red, linear), $I_{th}$ is 138.9 mW cm$^{-2}$. **d** The upconversion quantum yield and upconversion brightness of TTA-UCNP with different 650 nm incident power densities. $c$(glucose) = 5 mg mL$^{-1}$, $c$(GOX) = 32.5 μg mL$^{-1}$.

the intermolecular triplet energy transfer quantum yield from PdTPBP to perylene in the TTA-UCNP is determined to be 49.3%[43]. Moreover, the upconversion emission for the TTA-UCNP in PBS is power-dependent with a threshold intensity ($I_{th}$) of 138.9 mW cm$^{-2}$. Below 138.9 mW cm$^{-2}$, the upconversion emission intensity ($I_{UC}$) and the excitation intensity ($I_{ex}$) are in a quadratic relationship. When the $I_{ex}$ exceeds the threshold intensity, $I_{UC}$ and $I_{ex}$ are linear (Fig. 2c). The upconversion quantum yield ($\Phi_{UC}$) and respective brightness ($\eta = \varepsilon \times \Phi_{UC}$) of the TTA-UCNP in the presence of GOX and glucose were also measured and calculated. Below $I_{th}$, as the value of $I_{ex}$ increases, both the upconversion performance and the brightness increase. Beyond $I_{th}$, both two values reach a plateau ($\Phi_{UC}$ = 0.9% and $\eta$ = 1001) (Fig. 2d). We would like to note that, under the low-power excitation light, the TTA-UC is determined by an intermolecular collision between triplet excited states of annihilators ($^3$[An]$^*$), which is a bimolecular process. Thus, the power of the excitation light and the upconversion intensity have a quadratic relationship. In contrast, in high power excitation light, the intensity of TTA-UC typically depends on the radiation transition of the annihilator from the singlet excited state ($^1$[An]$^*$) to the ground state, which is a single molecule process[44]. Therefore, when the intensity of the excitation light is higher than $I_{th}$, the power of the excitation light and the upconversion intensity have a linear relationship and the TTA-UC quantum efficiency plateaus.

Since the phosphorescence of PdTPBP is sensitive to oxygen (Supplementary Fig. 10a)[10], we also prepared a nanoparticle with only PdTPBP that is coated by PAA-OA (PdTPBP NP) as a reference to further explore the mechanism of oxygen-clearance. Only weak phosphorescence was observed at 796 nm for PdTPBP NP in the absence of glucose and GOX (Supplementary Fig. 10b). This suggests that the PAA-OA polymer cannot solve the oxygen quenching problem per se. In addition, in the presence of only

GOX or only glucose, the phosphorescence of the PdTPBP NP is also not enhanced, indicating that only GOX or glucose cannot clear oxygen in an aqueous solution. In contrast, the phosphorescence of PdTPBP was effectively turned on with both GOX and glucose, indicating that oxygen consumption by GOX-catalyzed glucose oxidation took place (Supplementary Fig. 10b). Moreover, in the presence of GOX and glucose, we observed that the characteristic phosphorescence intensity of PdTPBP in TTA-UCNP is significantly lower than that in PdTPBP NP (Supplementary Fig. 11). This also supports the efficient TTET from PdTPBP to perylene inside TTA-UCNP[43].

Moreover, we further studied the influence of GOX concentration on the TTA-UC by detecting the kinetic rates of TTA-UC emission intensity for these TTA-UCNP under 650 nm illumination. This was done with different concentrations of GOX and the fixed concentration of glucose (5 mg mL$^{-1}$). A higher concentration of GOX was found to lead to a decreased upconversion half response time ($t_{1/2}$). For example, $t_{1/2}$ was shortened from 142 s to 20 s when the GOX concentration increased from 1.3 μg mL$^{-1}$ to 32.5 μg mL$^{-1}$ (Supplementary Fig. 12a). The fitting curve of $t_{1/2}$ and the concentration of GOX displayed exponential relation (Supplementary Fig. 12b), revealing that the enhancement of TTA-UC emission depends on the rate of oxygen consumption via the GOX catalyzed glucose oxidation reaction. The abovementioned results not only further verified that oxygen in the solution was consumed by the GOX-catalyzed glucose oxidation reaction, leading to upconversion emission "turn on", it also demonstrated that the upconversion response time was actually tunable by controlling the concentration of GOX in the TTA-UCNP aqueous solution.

**TTA-UCNP for glucose sensing.** Next, we set out to explore the feasibility of using this TTA-UC enhancement strategy to sense

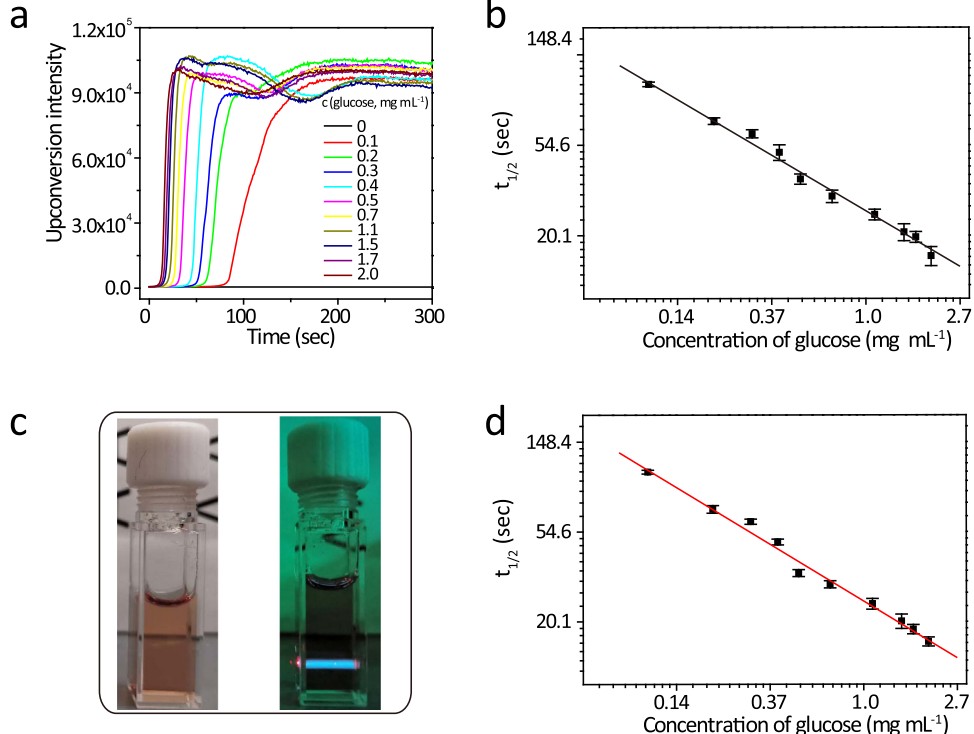

**Fig. 3 TTA-UNCP for efficient sensing of glucose. a** The response times of the mixture of TTA-UCNP (1 mg mL$^{-1}$) and GOX (10 µg mL$^{-1}$) with different concentrations of glucose, $\lambda_{ex}$ = 650 nm, 100 mW cm$^{-2}$. **b** In–In coordination fitting of glucose concentration and half response times in deionized water, slope = −0.62783, $R$ = 0.9995, $n$ = 3 means that each experiment is repeated three times independently, the error bar represents the mean of the three times ± standard deviation (SD). **c** The TTA-UCNP in the cell medium of 20% FBS, phenol red, and GOX (10 µg mL$^{-1}$) in the absence (left) and in the presence (right) of light, $\lambda_{ex}$ = 650 nm. **d** In–In coordination fitting of glucose concentration and half response times of TTA-upconversion in TTA-UCNP in cell culture medium including 20% FBS, phenol red and GOX (10 µg mL$^{-1}$), slope = −0.6333, $R$ = 0.994, $n$ = 3 means that each experiment is repeated three times independently, the error bar represents the mean of the three times ± standard deviation (SD).

glucose in an aqueous solution. D-glucose has played key roles in a wide variety of biological processes in living organisms[45]. An abnormal level of glucose in body fluid has been linked to several life-threatening diseases, such as diabetes[45], nephritis[46], as well as liver damage[47]. Therefore, glucose sensing has considerable importance in the biomedical field[48]. Prior to our study, the fluorescence glucose sensors were developed according to the capability of organic boronic acids to act as molecular receptors for saccharides, especially glucose[49]. Yet, the affinity of boronic acid towards glucose is not optimal and its glucose selectivity is not specific[50]. Meanwhile, in such a strategy, the conjugated reporter chromophores, such as anthracene, phenanthrene, and naphthalene, are typically involved in short-wavelength light excitation and emission (ultraviolet or blue light), the latter of which suffers from autofluorescence background from the biological fluids[51]. In addition, fluorescent probes or nanoparticles (gold or silver) were also used to indirectly estimate the amount of glucose by responding to $H_2O_2$ or protons (the products of the GOX catalytic glucose oxidation reactions)[50]. However, almost all these reported optical glucose sensors are based on changes in "downconverting" fluorescence or absorbance. Therefore, background interference remains and greatly reduces the specificity and sensitivity of glucose[50,51]. Since upconversion materials have opposite optical profiles compared to conventional fluorescence chromophores, we envisioned that unique GOX mediates TTA-UC enhancement can be used as a background-free glucose sensor[52–54].

As has been shown in the above-mentioned experimental results (Supplementary Fig. 12), GOX concentrations dictate kinetic rates of the recovery of TTA-UC emission in the presence

of glucose. Thus, for the purpose of glucose sensors, since the lower or higher end concentrations of GOX can lead to overly slow or fast readouts and subsequent potential experimental inconvenience and errors, we chose to use GOX of 10 µg mL$^{-1}$ to demonstrate the glucose-sensing ability of the TTA-UCNP. Glucose at different concentrations was added to a solution of GOX (10 µg mL$^{-1}$) and TTA-UCNP (1 mg mL$^{-1}$). We observed that the response times for the TTA-UCNP to approach its maximal upconversion intensity are dependent on the glucose concentration. More specifically, the half response time is found to be linearly decreased with the increased concentration of glucose in a log scale manner (Fig. 3). The limit of detection for such glucose sensing is deduced to be 0.06 mg mL$^{-1}$. In addition, by linear extrapolating the fitted straight line (Fig. 3b), this concentration of detection is observed to be up to 3.0 mg mL$^{-1}$. This concentration range (0.06–3 mg mL$^{-1}$) is wider than the requirement for clinical fasting blood glucose (0.7–1.1 mg mL$^{-1}$) testing. Furthermore, our TTA-UCNP glucose sensors are found to be highly selective for glucose (Supplementary Fig. 13). In this regard, we also tested a wide variety of other prevalent biological saccharides and polyhydric alcohols, such as fructose, 2-deoxy-d-ribose, sucrose, maltose, D-lactose, and D-(+)-raffinose, sodium D-gluconic acid, glycerol, D-mannitol, and glucuronolactone (Supplementary Fig. 13). As a result, insignificant upconversion emissions were observed with these compounds. The upconversion emission response was only observed with glucose. Moreover, in the presence of 5%, 10%, and 20% FBS respectively, there is no obvious change for TTA-upconversion intensity, suggesting that serum proteins have not significantly affect TTA-UCNP (Supplementary Fig. 14). In addition, in order to explore whether

our method can be interfered with by the autofluorescence background, we conducted the tests in the colored cell culture containing 20% FBS, phenol red, and GOX (10 μg mL$^{-1}$). With the addition of glucose (2 mg mL$^{-1}$), obvious TTA-upconversion was observed (Fig. 3c, right). Then, under this condition, we quantitatively measured the half response time of TTA-upconversion at different concentrations of glucose (Supplementary Fig. 15). We fitted the concentration of glucose and half response time in In–In coordination (Fig. 3d) and then compared it to Fig. 3b (in deionized water, slope = −0.627), observing an insignificant change in slope (−0.633), which suggested that the TTA-UCNP can still accurately measure the glucose concentration with autofluorescence background. Compared to the typical examples of pre-existing glucose quantitative analysis methods (Stokes-emission-based techniques) (Supplementary Table 1), our method, not only has a low detection line but also has a wider detection range.

Moreover, we would like to note that in previous studies, it was necessary to embed the chromophores, such as Ru, Pd, Pt complexes, and GOX, in a polymer or hydrogel in order to make the device measure the glucose content[50]. In contrast, our TTA-UCNP can be directly used to measure the glucose in an aqueous solution without the requirement of a complicated device preparation process. For the purpose of direct comparison, as an example, we also head-to-head compared our system with a well-established Ru(bpy)$_3$[55,56] in a nanoparticle formation to measure the glucose in the aqueous solution. In this controlled study, Ru NPs were constructed similar to the protocol with TTA-UCNP. As shown in Supplementary Fig. 16, in the presence of glucose and GOX, the phosphorescence intensity of Ru(bpy)$_3$ increases by 20% in DI water. However, such Ru NPs are not sensitive enough to quantitatively analyze the concentration of glucose below 1.6 mM, which can be well detected in our study. Moreover, in the presence of interference by the cell culture medium, the performance of such Ru(bpy)$_3$ is even worse. Only a 10% phosphorescence increase was observed in the presence of glucose and GOX and the detection was significantly impaired by the background.

**TTA-UCNP for the measurement of invertase activity**. Next, we sought to explore the feasibility to use our TTA-UCNP in conjugation with GOX as sensors in order to measure the activity of enzymes related to glucose metabolism. As a proof-of-principle, we studied the activity of the invertase (β-fructofuranosidase) using our TTA-UCNP sensor. Invertase is an important enzyme that catalyzes the hydrolysis of sucrose to produce D-glucose and D-fructose. These enzymes broadly exist in plants[57], microorganisms[58], and humans[59]. In particular, the activity of such an enzyme must be routinely measured in the many areas of biology[60,61], and bioengineering[62]. To date, invertase activity is determined by a couple of enzyme assays in which invertase cleaves sucrose to glucose and fructose, resulting in a colorimetric product, proportional to the invertase activity present[63]. However, these existing methods are based on additional colored indicators, such as oxiRed probe (absorbance at 570 nm) to determine invertase activity[64]. Moreover, there are typically biological pigments in the samples, such as chlorophyll, carotenes, and phytochromes, which have significant interference when measuring the activity of invertase. Contrary to these existing methods, the unique TTA-UC properties in our method are expected to overcome the problematic background interference and to minimize cross-talking with auto-fluorescence from the samples.

In our experiment (Fig. 4a), the sucrose and invertase were incubated at room temperature for 5 min. Then, the invertase

was deactivated in boiling water for 10 min. When the solution cooled to room temperature, it was added with the TTA-UCNP (1 mg mL$^{-1}$) and GOX (10 μg mL$^{-1}$) (Fig. 4a). As shown in Fig. 4b, we observed that the enzymatic reaction of the invertase can effectively turn on the original latent upconversion emissions of the TTA-UCNP. The higher concentration of invertase is found to lead to a quicker TTA upconversion response. Based on the results that can be seen in Fig. 3b, we calculated the concentrations of an intermediate product, glucose, and then converted them to the invertase activity. As shown in Fig. 4c, the invertase activity has a good linear relationship with half response time of TTA upconversion in In–In coordinates. The minimum detection limit is calculated to be 0.01 unit, which outperformed that of the commercial products (0.02 unit)[64]. These results show that our TTA-UCNP biosensors can also be extended to measure enzymatic activity that is related to glucose metabolism.

## Discussion

In summary, we developed a straightforward strategy to overcome the oxygen quenching problem to turn on the upconversion emissions of TTA-UCNP in the aqueous solution. This approach is based on the simple GOX catalytic reaction, which directly exhausts and depletes the oxygen in the aqueous solution. Moreover, we demonstrated this method can be extended as highly selective glucose sensors and to detect enzymes linked to glucose metabolism. To the best of our knowledge, this strategy is the first demonstration of the long-thought-after TTA-UCNP based solution as biological sensors. Therefore, our method should pave the way to use TTA-UCNP in a wide variety of photonic and biophotonic applications that are currently limited by the existing oxygen quenching problem.

## Methods

**Chemicals**. Perylene, octadecylamine, poly (isobutylene-alt-maleic anhydride) $M_w$ = 6000, tetrahydrofuran (THF), $N$, $N$-dimethylformamide (DMF), glucose, GOX, and invertase were purchased from Sigma-Aldrich (St. Louis, MO, USA). Meso-tetraphenyl-tetrabenzoporphine palladium complex (PdTPBP) was purchased from Fisher Scientific. Ultrapure water was prepared by using a Millipore Simplicity System (Millipore, Bedford, USA). All of the above-mentioned chemicals were used as received without further purification.

**Characterization**. UV–Vis spectra were recorded via an Agilent Cary-5 spectrophotometer. Steady-state fluorescence spectra were measured with a Hitachi F-7000 fluorescence spectrometer. The morphology of the TTA-UC nanoparticle was characterized by using a JEOL JEM-200CX TEM operated at 80 kV. The sample for TEM measurement was prepared by dropping the solution onto a carbon-coated copper grid after negative staining with 10.0% (w/v) sodium phosphotungstic acid. The particle size and size distribution of the TTA-UCNP in an aqueous solution was measured by using dynamic light scattering (DLS) via a Malvern Zetasizer Nano ZS. The original data is re-drawn and analyzed on Origin Pro 8.

**PAA-OA synthesis**. PAA-OA synthesis followed a similar method in the literature[65]. Poly (isobutylene-alt-maleic anhydride) ($M_w$ = 6000) (1.85 g) and octadecylamine (1.6 g) were dissolved in dry THF (100 mL). They were then left to react for 24 h at 70 °C. The solvent was evaporated under reduced pressure to attain the PAA-OA product.

**Synthesis of TTA-UCNP**. The TTA-UCNP was prepared via self-assembly of the sensitizer (PdTPBP), the annihilator (perylene) with PAA-OA. Briefly, PdTPBP (50 μM), perylene (1 mM), and PAA-OA (75 mg) were dissolved in 5 mL THF and followed by the addition of 5 mL 50 mM sodium borate buffer (with a pH of 9.16). The mixture solution was stirred at 40 °C for 2 h and the volume of the solution is reduced due to the evaporation of the THF. The reaction mixture was then cooled to room temperature and dialyzed in deionized water for 24 h. After that, we concentrated the nanoparticles to 5 mL via an ultracentrifuge tube (cut off $M_w$ = 3000). The TTA-UCNP solution was then stored at 4 °C. The PdTPBP nanoparticles (PdTPBP NP) were prepared by a similar method. PdTPBP (50 μM) and PAA-OA (75 mg) were used for preparing PdTPBP NP.

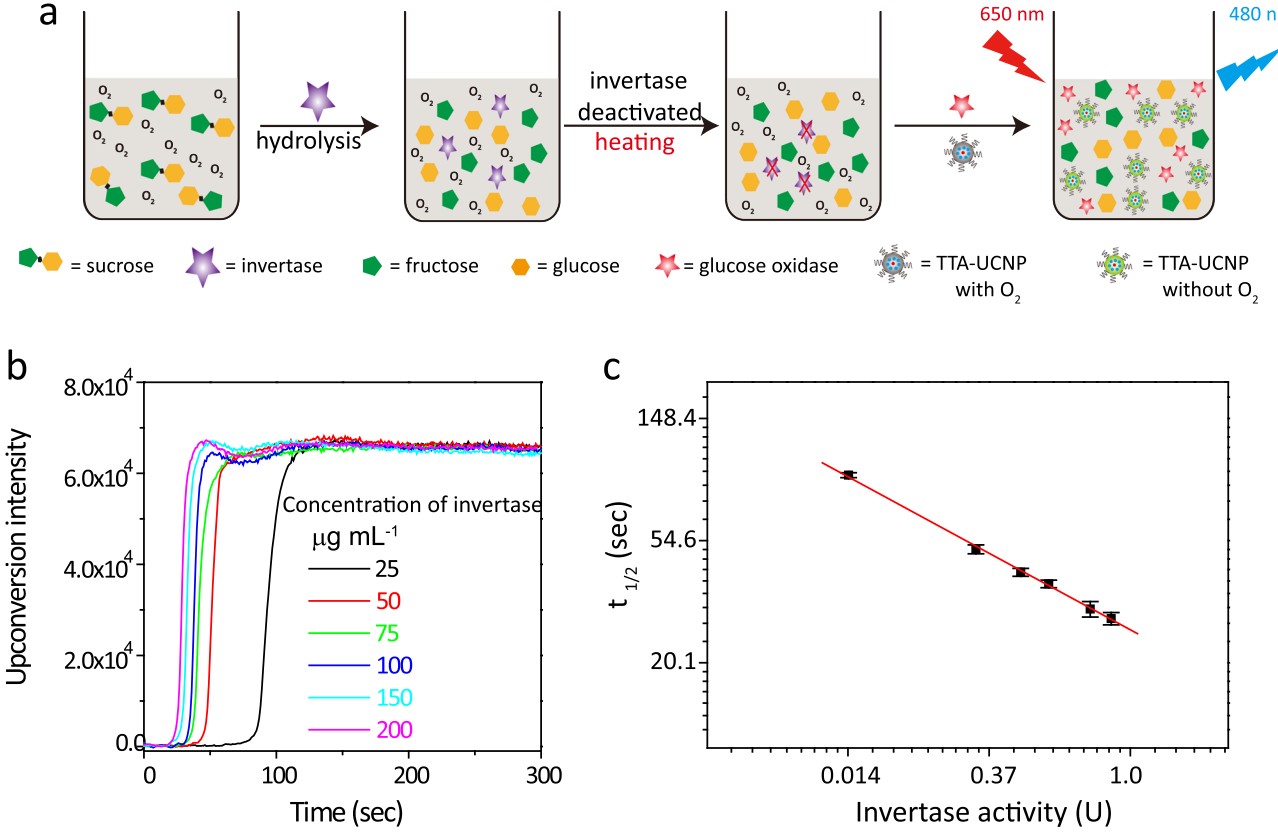

**Fig. 4 TTA-UCNP measures the activity of invertase. a** Schematic illustration of the mechanism of TTA-UCNP as a biosensor for invertase enzymatic reactions. **b** The invertase concentration-dependent TTA-upconversion turn-on response. $\lambda_{ex} = 650$ nm, 100 mW cm$^{-2}$. **c** In–In coordination fitted linear relationship between the concentration of invertase and TTA-upconversion half response time (slope $= -0.625$, $R = 0.999$), $n = 3$ means that each experiment is repeated three times independently, the error bar represents the mean of the three times ± standard deviation (SD).

**The TTA upconversion measurement in toluene**. A continuous diode-pumped solid-state laser (650 nm) was used as the excitation source for the upconversion measurement. For the upconversion measurements, the mixed solution of PdTPBP and perylene was degassed with Argon for 15 min. The solution was then excited with the laser, and the corrected upconversion spectrum was recorded with a HORIBA spectrofluorometer.

**The measurement and calculation of upconversion quantum yield ($\Phi_{uc}$)**. The $\Phi_{uc}$ was calculated by an established method[1,11]. The methylene blue in methanol with fluorescence quantum yield ($\Phi_f = 3\%$) was used as the reference. The upconversion quantum yield was calculated with Eq. (1), where $\Phi_{uc}$, $A_{std}$, $I_{std}$, and $\eta_{std}$ represent the upconversion quantum yield, the absorbance of the reference, the integrated photoluminescence intensity of the reference, and the refractive index of the solvents:

$$\Phi_{uc} = 2 \times \Phi_f \times \left(\frac{A_{std}}{A_{unk}}\right) \times \left(\frac{I_{unk}}{I_{std}}\right) \times \left(\frac{\eta_{unk}}{\eta_{std}}\right)^2 \quad (1)$$

where $\Phi_{uc}$, $A_{std}$, $I_{std}$, and $\eta_{std}$ represent the upconversion quantum yield, the absorbance of the reference, the integrated photoluminescence intensity of the reference, and the refractive index of the solvents.

**The entrapment efficiency of dyes in nanoparticles**. The entrapment efficiency of dyes was measured via a reported protocol[66,67]. We measured the absorbance of PdTPBP or perylene in dichloromethane ($A_0$) before synthesis of the TTA-UCNP. We added the dichloromethane to extract the PdTPBP and the perylene from the final TTA-UCNP product to attain absorbance ($A$) for PdTPBP or perylene in the TTA-UCNP. Then the entrapment efficiency of PdTPBP or perylene was calculated according to Eq. (2), which has been used similarly in the literature[66,67].

$$\text{Entrapment efficiency}(\%) = (A)/(A_0) \times 100 \quad (2)$$

**TTA upconversion measurement of the TTA-UCNP in PBS buffer**. A continuous diode-pumped solid-state laser (650 nm) was used as the excitation source for the upconversion. For the upconversion measurements, the TTA-UCNP was irradiated by 650 nm laser under different conditions (in the absence of glucose and GOX, in the presence of glucose and GOX, in the presence of glucose or GOX

alone). The upconversion emission spectra of TTA-UCNP were recorded with a HORIBA spectrofluorometer after the upconversion emission intensities of the TTA-UCNP were stabilized. The $\Phi_{uc}$ of the TTA-UCNP in PBS buffer was calculated according to the above-mentioned equation. The TTA upconversion brightness is then calculated according to Eq. (3) that has been used similarly in the literature[68]

$$\eta = \Phi_{uc} \times \varepsilon \quad (3)$$

where $\eta$ is the TTA upconversion brightness; $\Phi_{uc}$ is the TTA upconversion quantum yield of TTA-UCNP; $\varepsilon$ is the molar extinction absorption coefficient of PdTPBP.

**TTET efficiency measurements[43]**. The PdTPBP-to-perylene TTET quantum yield in TTA-UCNP was calculated from the measurements of the PdTPBP phosphorescence in the presence of perylene ($I$, TTA-UCNP) and in the absence of the perylene ($I_0$, PdTPBP NP), using Eq. (4) that has been used in the literature[43].

$$\Phi_{(TTET)} = 1 - I/I_0 \quad (4)$$

**Calculation of the detection limit of glucose**. The detection limit of glucose was calculated based on a reported method[69]. A linear relation exists between the half response time ($y$, $t_{1/2}$) and the concentration of glucose in In–In coordination (Fig. 3c). This linear relationship is fitted as in Eq. (5)

$$y = -0.628x + 3.27 \quad (5)$$

Based on the theory of Martins and Naes[70], the detection limit can be derived from the signal processing performance, as in Eq. (6) described below

$$V_{x^2} = \sum(y_i - y)^2 \quad (6)$$

where $y_i$ is the average value from the calculation and $y$ is the measured data points.

The background of root-mean-square (rms) $rms_{noise}$ is calculated using Eq. (7)

$$rms_{noise} = \sqrt[2]{\frac{V_{x^2}}{N}} \quad (7)$$

where $N$ is the number of data points ($N = 3$) used for the average value.

Then the detection limit of the glucose is calculated to be 0.06 mg mL$^{-1}$ using Eq. (8).

$$\text{The detection limit} = 3\times \text{rms}_{\text{noise}}/(\text{slope of the In} - \text{In coordination line}) \quad (8)$$

**The measurement of invertase activity**. First, we measured the half response time of TTA-UCNP (1 mg mL$^{-1}$) in conjunction with GOX (10 μg mL$^{-1}$) at different concentrations of glucose (0, 0.1, 0.2, 0.3, 0.4, 0.5, 0.7, 1.1, 1.5, and 2.0 mg mL$^{-1}$) to attain the standard curve in deionized water. The results were shown in Fig. 4b. Next, sucrose (100 mg mL$^{-1}$) and different concentrations of invertase were incubated at room temperature for 5 min. After that, the invertase was deactivated in boiling water (100 °C) for 10 min. When the solution further cooled down to room temperature (25 °C), TTA-UCNP (1 mg mL$^{-1}$) and GOX (10 μg mL$^{-1}$) were added. We then measured the half response times of the TTA-upconversion. According to the standard curve (Fig. 4b), we can calculate and deduce the glucose concentration in the solution. Since in the reaction a mole of sucrose is hydrolyzed to produce mole glucose and mole fructose, the amount of sucrose is equal to that of glucose. Therefore, we can calculate the amount of hydrolyzed sucrose using the amount of glucose. Then the actual invertase activity ($U$) can be calculated using Eq. (9) that has been used in a commercial protocol[64].

$$\text{Invertase activity}(U) = \text{Amount of sucrose}\,(\mu mol)/5\,\text{min} \quad (9)$$

**Calculation of the detection limit of invertase**. The process for the calculation of the detection limit of invertase is similar to the above-mentioned calculation of the detection limit of glucose. A linear relation exists between the half response time ($y$, $t_{1/2}$) and the activity of invertase ($x$, unit) in In–In coordination (Fig. 4c). This linear relation is fitted as Eq. (10):

$$y = -0.625x + 3.28 \quad (10)$$

Then using the above-mentioned equations for glucose detection limit, the invertase detection limit is calculated to be 0.01 unit.

**Reporting Summary**. Further information on research design is available in the Nature Research Reporting Summary linked to this article.

## Data availability

Data supporting the findings of this study are available within the paper and its Supplementary Information files. The source data underlying Figs. 2–4 and Supplementary Information Figs. 1–5, 8, and 12–16 are provided as a Source Data file. The data that support the findings of this study are available from the corresponding author on request.

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

## Acknowledgements

Financial support was provided by the National Institutes of Health (NIH) awards R01CA232017 and R21GM125632.

## Author contributions

L.H. conducted the majority of experiments, ran most of the data analysis, and wrote the manuscript. T.L. is responsible for preparing nanoparticles. K.H. is responsible for the characterization of nanoparticles including transmission electron microscopy and hydrodynamic particle size testing. G.H. conceived and supervised the project and experiments and wrote the manuscript.

## Competing interests

Authors declare the following competing interests: L.H. and G.H. have a pending patent application are related to this study. T.L. and K.H. declare no competing interests.
