## [Peer Review File · Nature Communications]

REVIEWER COMMENTS

Reviewer #1 (Remarks to the Author):

The authors report a simple enzymatic strategy to overcome oxygen-induced TTA-UC quenching. The nanoparticles based on enzymatic TTA-UC enhancement strategy can also detect glucose and measure the activity of enzymes related to glucose metabolism in a highly specific, and sensitive. This work is useful for the development of efficient TTA-UC nanoparticles in aqueous solution and thus facilitates their biological applications. Breaking oxygen quenching and developing new applications, this is a very good example to show the potential of TTA-UC. However, the manuscript is just not suitable for publication at present. Some concerns and suggestions about the manuscript are shown below.

1. The manuscript should be checked carefully. Obviously, there are a lot of problems in the manuscript in terms of the non-uniform formats, and mistakes. Here are some representative examples mainly about spelling mistakes.

(i) Page 14, spelling mistake. “In In-In coordination, the concentration of glucose and half response time was fitted (Fig 2e).”

(ii) Page 18. “(11) Huang, L.; Zhao, Y.; Zhang, H.; Huang, K.; Yang, J.; Han, G. *Angew. Chem. Int. Ed.* 2017, 56,14400–14404.”

(iii) Page 11. “This also supports the efficient triplet-triplet energy transfer from PdTPBP to perylene inside TTA-UCNP43”

(iv) Page 7. “GOX is biocompatible and has been widely used in medicine³⁹ and biology⁴⁰ 41”

(v) Page 5. “However, the existing silica TTA-UCNP cannot further block the oxygen quenching and thus has suboptimal upconversion performance and the size of the nanoparticles is very large, typically over 200 nm¹⁰, 19.”

(vi) Page 4. “To date, a few TTA-UC photosensitizers and annihilators pairs with solid upconversion quantum yield have been reported to be in deaerated organic solvents³ 14, 15”

(vii) Page 15. “These enzymes broadly exist in plants⁵⁴, microorganisms ⁵⁵ and humans ⁵⁶. In particular, the activity of such an enzyme must be routinely measured in the many areas of biology, ⁵⁷, ⁵⁸ and biomass engineering⁵⁹.”

2. The research work developed a straightforward strategy to overcome the oxygen quenching problem to turn on the upconversion emission. The approach is based on the glucose oxidase catalytic reaction, which directly exhaust and deplete the oxygen in the aqueous solution. The TTA-UC enhancement and detection application are both on the basis of the oxygen-consuming strategy. In this respect, this is very similar to a previous literature of *J. Mater. Chem. C*, 2016, 4, 9986-9992, where the Time–oxygen & light indicating via photooxidation mediated upconversion was reported.

3. The “background-free manner” is not sufficiently demonstrated in the determination of invertase activity applications. As the authors declare in Page 15, the existing colorimetric method for the determination of invertase activity has significant interference, especially colored substrates. Maybe, some brief connections or explanations are needed between the “colored substrates” and “auto fluorescence”.

The results in this manuscript show that TTA-UC biosensors can be extended to measure enzymatic activity that is related to glucose metabolism. Particularly, the performance is better than the commercial products.

In brief, this work is very important while the manuscript should be revised to definitely address the issues above. Therefore, I recommend a minor revision.

Reviewer #2 (Remarks to the Author):

This paper reports the TTA-UC system, which is realized by glucose oxidase (GOX) catalyzed glucose oxidation, and subsequent oxygen depletion. The synthesized TTA-UC nanoparticle (TTA-UCNP) is then used to detect glucose and to measure the enzymatic activity in aqueous solution. The strength of this paper lies in the contrarian idea of using TTA-UCNP as a sensor by using the oxygen quenching, which has been considered as a hurdle in TTA-UC for a long time. It is also meaningful that this is the first proof-of-concept demonstration of using TTA-UCNP as a biological sensor to detect glucose and to measure enzymatic activity with minimal background noise such as autofluorescence. All synthesis, experiments, and photochemical characterization were also performed carefully with the appropriate control experiments. Overall, this research was carried out to a very high standard, and also addressed a new application of TTA-UC as a biosensor. I recommend publication of this manuscript after minor revision as follows:

Page 11~14, Fig. 2. The authors pointed out the shortcomings of the existing methods and describe the advantages of glucose sensing using TTA-UCNP (i.e. selectivity and low background autofluorescence). But I think it is necessary to compare these results with that obtained from the pre-existing glucose quantitative analysis methods (stokes-emission-based techniques) to highlight this strategy.

Page 9. Here, the authors analyzed and reported quantum efficiency and power dependency of the TTA-UCNP that they synthesized. However, to this reviewers, it seems it is only a simple description of the well-known characteristics of TTA-UC (i.e. quadratic to linear power dependency, QY plateau at high laser intensity). In other words, the characteristics of the material reported in this paper are not clearly presented. Is the quantum efficiency of this material higher than that of the material reported in other papers? Or comparable? Is the threshold intensity (I_{th}) lower than previously reported value? In addition, it is needed to briefly describe why this tendency (quadratic to linear power dependency, QY plateau at high laser intensity) appears.

Fig 2a shows a typical laser setup that analyzes photoluminescence and does not contain something new. It would be better to move to SI.

Typo in Fig. 3a deactivaed □ deactivated

Point-to-point Responses to REVIEWER COMMENTS

Reviewer #1 (Remarks to the Author):

The authors report a simple enzymatic strategy to overcome oxygen-induced TTA-UC quenching. The nanoparticles based on enzymatic TTA-UC enhancement strategy can also detect glucose and measure the activity of enzymes related to glucose metabolism in a highly specific, and sensitive. This work is useful for the development of efficient TTA-UC nanoparticles in aqueous solution and thus facilitates their biological applications. Breaking oxygen quenching and developing new applications, this is a very good example to show the potential of TTA-UC. However, the manuscript is just not suitable for publication at present. Some concerns and suggestions about the manuscript are shown below.

Reply: we appreciate the Reviewer's acknowledgment of the significance and novelty of this paper, as well as your positive comments and informative suggestions. We have provided a point-to-point response to the questions kindly raised by the Reviewer.

1. The manuscript should be checked carefully. Obviously, there are a lot of problems in the manuscript in terms of the non-uniform formats, and mistakes. Here are some representative examples mainly about spelling mistakes.

(i) Page 14, spelling mistake. "In In-In coordination, the concentration of glucose and half response time was fitted (Fig 2e)."

(ii) Page 18. "(11) Huang, L.; Zhao, Y.; Zhang, H.; Huang, K.; Yang, J.; Han, G. *Angew. Chem. Int. Ed.* 2017, 56,14400–14404."

(iii) Page 11. "This also supports the efficient triplet-triplet energy transfer from PdTPBP to perylene inside TTA-UCNP43"

(iv) Page 7. "GOX is biocompatible and has been widely used in medicine³⁹ and biology^{40, 41}"

(v) Page 5. "However, the existing silica TTA-UCNP cannot further block the oxygen quenching and thus has suboptimal upconversion performance and the size of the nanoparticles is very large, typically over 200 nm^{10, 19}."

(vi) Page 4. "To date, a few TTA-UC photosensitizers and annihilators pairs with solid upconversion quantum yield have been reported to be in deaerated organic solvents^{3, 14, 15}"

(vii) Page 15. "These enzymes broadly exist in plants⁵⁴, microorganisms ⁵⁵ and humans ⁵⁶. In particular, the activity of such an enzyme must be routinely measured in the many areas of biology, ^{57, 58} and biomass engineering⁵⁹."

Reply: Thank you very much for so carefully bringing these errors to our attention. We have revised these points based on your important input. These changes have been highlighted in yellow.

2. The research work developed a straightforward strategy to overcome the oxygen quenching problem to turn on the upconversion emission. The approach is based on the glucose oxidase catalytic reaction, which directly exhaust and deplete the oxygen in the aqueous solution. The TTA-UC enhancement and detection application are both on the basis of the oxygen-consuming strategy. In this respect, this is very similar to a previous literature of *J. Mater. Chem. C*, 2016, 4, 9986-9992, where the Time–oxygen & light indicating via photooxidation mediated upconversion was reported.

Reply: We thank the Reviewer for this comment. We would like to clarify this point. In this reference paper kindly noted by the Reviewer (*J. Mater. Chem. C*, 2016, 4, 9986-9992), the reported method was able to be used only in organic solvent (toluene). Thus, prior to our study, breaking oxygen quenching in biological related aqueous systems has still been extremely challenging. To overcome this key, urgent problem, we proposed the use of the glucose oxidase catalytic reaction, which is biocompatible, in our study to reduce oxygen mediated TTA-UC quenching in water. It is our hope that the Reviewer can concur with this clear difference between two papers. Nevertheless, we respect the Reviewer's opinion here and have included the referenced paper in our references.

3. The “background-free manner” is not sufficiently demonstrated in the determination of invertase activity applications. As the authors declare in Page 15, the existing colorimetric method for the determination of invertase activity has significant interference, especially colored substrates. Maybe, some brief connections or explanations are needed between the “colored substrates” and “auto fluorescence”. The results in this manuscript show that TTA-UC biosensors can be extended to measure enzymatic activity that is related to glucose metabolism. Particularly, the performance is better than the commercial products.

Reply: We really appreciate the Reviewer’s kind words about our results, as well as this kind of advice in regard to the respective discussion. Per your kind suggestion, we have revised the manuscript and made the connections and explanations to make our point clearer to the readers.

In brief, this work is very important while the manuscript should be revised to definitely address the issues above. Therefore, I recommend a minor revision.

Reply: We are grateful to the Reviewer for your positive comments and kind words. We hope that this revision has addressed your questions properly.

Reviewer #2

Title: Breaking Oxygen Quenching: Enzymatic Enhancing Triplet-triplet Annihilation Upconversion in Aqueous Solutions for Back-ground free Self- Standing Biological Sensing

Authors: Ling Huang, Timmy Le, Kai Huang, Gang Han

This paper reports the TTA-UC system, which is realized by glucose oxidase (GOX) catalyzed glucose oxidation, and subsequent oxygen depletion. The synthesized TTA-UC nanoparticle (TTA-UCNP) is then used to detect glucose and to measure the enzymatic activity in aqueous solution. The strength of this paper lies in the contrarian idea of using TTA-UCNP as a sensor by using the oxygen quenching, which has been considered as a hurdle in TTA-UC for a long time. It is also meaningful that this is the first proof-of-concept demonstration of using TTA-UCNP

as a biological sensor to detect glucose and to measure enzymatic activity with minimal background noise such as autofluorescence. All synthesis, experiments, and photochemical characterization was also performed carefully with the appropriate control experiments. Overall, this research was carried out to a very high standard, and also addressed a new application of TTA-UC as a biosensor. I recommend publication of this manuscript after minor revision as follows:

Reply: We appreciate the Reviewers comments about the rigorous and high standards in regard to our research. We also thank you for your suggestions to help us improve our studies.

(1) Page 11~14, Fig. 2. The authors pointed out the shortcomings of the existing methods and describe the advantages of glucose sensing using TTA-UCNP (i.e. selectivity and low background autofluorescence). But I think it is necessary to compare these results with that obtained from the pre-existing glucose quantitative analysis methods (Stokes-emission-based techniques) to highlight this strategy.

Reply: Per your important suggestion, we have now compared our results with that obtained from the pre-existing glucose quantitative analysis methods (Stokes-emission-based techniques). (please refer to Table R1 or Table S1). Our experimental results show that our method, not only has a low detection line, it also has a wider detection range than these Stokes-emission-based techniques.

Moreover, we would like to note that in previous studies, it was necessary to embed the chromophores, such as Ru, Pd, Pt complexes and GOX, in a polymer or hydrogel in order to make the device measure the glucose content. In contrast, our TTA-UCNP can be directly used to measure the glucose in aqueous solution without the requirement of a complicated device preparation process. For the purpose of direct comparison, as an example, we also head-to-head compared our system with a well-established Ru(bpy)₃ ((*Anal. Chem.* 1990, 62, 2377-2380; *Anal. Chem.* 1995, 67, 3746-3752) in a nanoparticle formation to measure the glucose in the aqueous solution. In this controlled study, Ru NPs were constructed similar to the protocol with TTA-UCNPs. As shown in Fig R1 or Fig S16, in the presence of glucose and GOX, the phosphorescence intensity of Ru(bpy)₃ increases by 20% in DI water. However, such Ru NPs is not sensitive enough to quantitatively analyze the concentration of glucose below 1.6 mM, which can be well detected in our study. Moreover, in the

presence of interference by the cell culture medium, the performance of such Ru(bpy)₃ is even worse. Only a 10% phosphorescence increase was observed in the presence of glucose and GOX and the detection was significantly impaired by the background. Thank you again for your thoughtful suggestion. We have included the above discussion and new results to highlight the advantage of our method.

Table R1: Typical examples to determine glucose in the literature

Sensor	Excitation/Emission	AR: analytical range
GOx and probe Ru(phen)	exc./em. 488/610 nm	0.7–10 mM
GOx and Ru(phen)	exc./em. 468/570 nm	0.5–15 mM
EuTC and Hypan	exc./em. 400/616 nm	0.1–2 mM
Anthracene based bis-PhBA	exc./em. 370/423 nm	0.3–1 mM
TTA-UCNP	exc./em. 650/480 nm	0.33-17 mM

Fig R1. (a) UV-vis absorption spectra of Ru NPs in DI water; (b) Phosphorescence emission spectra of Ru NPs in different conditions in DI water, GOX (10 µg/mL),

glucose (3.0 mg/mL); (c) the response times of the mixture of Ru NPs and GOX (10 μ g/mL) in the different concentrations of glucose, $\lambda_{\text{ex}} = 460$ nm; (d) phosphorescence emission spectra of Ru NPs in different conditions in presence of cell culture medium, GOX (10 μ g/mL), glucose (3.0 mg/mL).

(2) Page 9. Here, the authors analyzed and reported quantum efficiency and power dependency of the TTA-UCNP that they synthesized. However, to this reviewer, it seems it is only a simple description of the well-known characteristics of TTA-UC (i.e. quadratic to linear power dependency, QY plateau at high laser intensity). In other words, the characteristics of the material reported in this paper are not clearly presented. Is the quantum efficiency of this material higher than that of the material reported in other papers? Or comparable? Is the threshold intensity (I_{th}) lower than previously reported value? In addition, it is needed to briefly describe why this tendency (quadratic to linear power dependency, QY plateau at high laser intensity) appears.

Reply: We are grateful to the Reviewer for this important suggestion. We would like to clarify that, to the best of our knowledge, long wavelength light mediated TTA-UCNP with a size that is below 50 nm has not been reported in aqueous solution. Thus, we are not able to directly compare our results with prior studies in regard to similar sized nanoparticles.

We used an established organic solvent based TTA pair (PdTPBP/perylene) to construct nanoparticles. The QY efficiency and threshold in deaerated toluene has previously been reported. These numbers are reconfirmed in our study to be 7.9% (QY) and 88.5 mW/cm² (TTA threshold intensity, I_{th}) in deaerated toluene.

With respect to the TTA-UC nanoparticles, we observed that the TTA-UC efficiency is 0.9% in aqueous solution and the I_{th} is 138.9 mW/cm². This difference in organic solvent and small aqueous soluble nanoparticles is likely to due to the well-known factors of the chromophore concentration quenching and water quenching (in the case of the absence of GOX). As you kindly mentioned, "the strength of this paper lies in the contrarian idea of using TTA-UCNPs as a sensor by using oxygen quenching." In the presence of the reaction of GOX and glucose, the reduced oxygen quenching and the accompanying QY increase was utilized as a sensor.

Per your latter suggestion, we have now also included a brief description, which follows, in order to make the point more clear to a general audience .

In general, under the low-power excitation light, the TTA-UC is determined by intermolecular collision between triplet excited state of annihilators ($^3[\text{An}]^*$), which is a bimolecular process. Thus, the power of the excitation light and the upconversion intensity have a quadratic relationship. In contrast, in high power excitation light, the intensity of TTA-UC typically depends on the radiation transition of the annihilator from the singlet excited state ($^1[\text{An}]^*$) to the ground state, which is a single molecule process (*Phys. Rev. B.* 78, 195112 (2008)). Therefore, when the intensity of the excitation light is higher than I_{th} , the power of the excitation light and the upconversion intensity have a linear relationship and the TTA-UC quantum efficiency plateaus.

(3) Fig 2a shows a typical laser setup that analyzes photoluminescence and does not contain something new. It would be better to move to SI.

Reply: Thank you for this valuable suggestion. we have now removed Fig 2a to SI (Scheme S1).

(4) Typo in Fig. 3a deactivaed \square deactivated

Reply: Thank you for pointing this error. We have now revised the mistake in main text.

REVIEWERS' COMMENTS

Reviewer #1 (Remarks to the Author):

The authors have revised the manuscript according to the reviewers' comments. Now it is acceptable.

Reviewer #2 (Remarks to the Author):

The authors have adequately addressed all the points I raised, providing a satisfying and insightful answer for each question. Therefore I recommend this manuscript for publication in Nature Communications.

AUTHOR'S RESPONSE TO REVIEWER COMMENTS

Reviewer #1 (Remarks to the Author):

The authors have revised the manuscript according to the reviewers' comments.

Now it is acceptable.

Reply: We are very grateful to the reviewers for reading our manuscript again and agreeing to accept it by Nature Communications.

Reviewer #2 (Remarks to the Author):

The authors have adequately addressed all the points I raised, providing a satisfying and insightful answer for each question. Therefore, I recommend this manuscript for publication in Nature Communications.

Reply: We are very grateful to the reviewers for being able to read our revised manuscript and agree to accept it by Nature Communications.